

# Metabolites profiling of Sapota fruit pulp *via* a multiplex approach of gas and ultra performance liquid chromatography/mass spectroscopy in relation to its lipase inhibition effect

Mohamed A. Farag[1], Nermin Ahmed Ragab[2] and Maii Abdelnaby Ismail Maamoun[2]

[1] Department of Pharmacognosy, Cairo University, Cairo, Egypt
[2] Department of Pharmacognosy, National Research Center, Giza, Egypt

## ABSTRACT

**Background**. Sapota, *Manilkara zapota* L., are tasty, juicy, and nutrient-rich fruits, and likewise used for several medicinal uses.

**Methods**. The current study represents an integrated metabolites profiling of sapota fruits pulp *via* GC/MS and UPLC/MS, alongside assessment of antioxidant capacity, pancreatic lipase (PL), and α-glucosidase enzymes inhibitory effects.

**Results**. GC/MS analysis of silylated primary polar metabolites led to the identification of 68 compounds belonging to sugars (74%), sugar acids (18.27%), and sugar alcohols (7%) mediating the fruit sweetness. Headspace SPME-GC/MS analysis led to the detection of 17 volatile compounds belonging to nitrogenous compounds (72%), ethers (7.8%), terpenes (7.6%), and aldehydes (5.8%). Non-polar metabolites profiling by HR-UPLC/MS/MS-based Global Natural Products Social (GNPS) molecular networking led to the assignment of 31 peaks, with several novel sphingolipids and fatty acyl amides reported for the first time. Total phenolic content was estimated at $6.79 \pm 0.12$ mg gallic acid equivalent/gram extract (GAE/g extract), but no flavonoids were detected. The antioxidant capacities of fruit were at $1.62 \pm 0.2$, $1.49 \pm 0.11$, and $3.58 \pm 0.14$ mg Trolox equivalent/gram extract (TE/g extract) *via* DPPH, ABTS, and FRAP assays, respectively. *In vitro* enzyme inhibition assays revealed a considerable pancreatic lipase inhibition effect ($IC_{50} = 2.2 \pm 0.25$ mg/mL), whereas no inhibitory effect towards α-glucosidase enzyme was detected. This study provides better insight into sapota fruit's flavor, nutritional, and secondary metabolites composition mediating for its sensory and health attributes.

## INTRODUCTION

One of the biggest issues today is obesity, WHO predicts that by 2025, around 167 million individuals will encounter health problems owing to obesity (*WHO, 2022*). Obesity affects all of the body's organs leading to a wide range of metabolic disorders, such as diabetes

Corresponding author
Mohamed A. Farag,
mohamed.farag@pharma.cu.edu.eg

and hyperlipidemia, especially in low and middle-income countries. Current therapeutic strategies to manage hyperglycemia and hyperlipidemia depend on the inhibition of digestive enzymes. Thereby, relying on pancreatic lipase inhibitors is a potential approach to reduce fat absorption and weight while inhibiting $\alpha$-glucosidase enzyme can lead to the slower digestion of carbohydrates and postprandial glucose control (*Prieto-Rodríguez et al., 2022*). Dietary antioxidants could potentially contribute to the prevention and management of obesity. These phytoconstituents have a regulatory effect on cellular processes involved in adipose tissue function and energy metabolism (*Almoraie & Shatwan, 2024*).

Several approved anti-obesity drugs have now been discontinued due to their adverse effects such as bloating, flatulence, oily stools, diarrhea, abdominal pain, and reduction in fat-soluble vitamin absorption (*Oshiomame Unuofin, Aderonke Otunola & Jide Afolayan, 2018*). Additionally, various oral anti-diabetic drugs are contraindicated to other medications (*Chaudhury et al., 2017*). Thus, it is crucial to find an anti-obesity medication from natural resources, especially with various natural products reported to reduce body weight and blood glucose levels.

Sapotaceae is a family of tropical, evergreen trees and shrubs that comprises more than 50 genera and 1,100 species. One of the most famous genera in the Sapotaceae family is *Manilkara* with *ca.* 80 species. The trees of *Manilkara zapota* (L.) Van Royen is the most extensively grown species (*The Plant List, 2010*; *Madani et al., 2018*).

*Manilkara zapota* (L.) fruits, also known as sapota, sapodilla, and chicozapote, are uniquely delicious, with a delicate, grainy feel and pleasant smell. The fruit comprises a wide array of nutrients, minerals, and polyphenols with diverse biological activities (*Siddiqui et al., 2014*). Unripe fruits are astringent in taste due to high levels of catechins, chlorogenic acid, gallotannins, and proanthocyanidins (*Ma et al., 2003*). Aside from phenolics, fruits are also rich in triterpenoids represented by β-amyrin-3-(3′-dimethyl) butyrate and lupeol-3-acetate (*Fayek et al., 2013*). The fruit aqueous extract exhibits potential anti-hypercholesterolemic, antihyperglycemic, and antioxidant effects (*Fayek et al., 2013*). Previous results revealed a decrease in body weight and a good influence on biomarkers, *viz*, insulin, glycemia, cholesterol, and triglycerides in animals treated with *M. zapota* fruit juice, highly indicating the antidiabetic and anti-obesity potentials of its fruit (*Barbalho et al., 2015*).

Metabolite profiling assesses the nutritional value and bioactive constituents of botanical samples mediating for their health benefits. The current study presents a multiplex approach employing gas chromatography coupled with mass spectrometry (GC/MS), and high-resolution ultra-high performance liquid chromatography coupled with tandem mass spectrometry (HR-UPLC/MS/MS) for profiling *M. zapota* (L.) fruit pulp targeting its aroma, non-volatile polar and non-polar metabolites to account for fruit sensory, nutritional, and health attributes. The aroma profile was assessed using solid phase micro-extraction (SPME), whereas primary polar metabolites *viz.* sugars were analyzed using GC/MS post-silylation. For large molecular weight non-polar metabolites analysis, HR-UPLC/MS/MS was employed aided by Global Natural Products Social (GNPS) molecular networking to assist in metabolites identification. In addition, total phenolics and flavonoids were determined for standardization, alongside lipase and $\alpha$-glucosidase

inhibition activities of fruit extract. The main goal of this study is the identification and development of natural curing agents for metabolic disorders.

## MATERIALS AND METHODS

### Plant material and extraction process

The fresh fruit pulp of sapota (*Manilkara zapota* L.) was collected from Haryana Agriculture University, Hisar, India in December 2022. The plant sample was identified by Dr. Rupesh Deshmuk, Central University of Haryana, India. Fruits were immediately lyophilized by Telstar Laboratory Freeze Dryers, peeled and the pulp was taken, treated with liquid nitrogen powdered using mortar and pestle, and stored in closed, air-tight bags at −20 °C until further analysis. The extraction process was carried out following the previously mentioned procedure (*El-Akad, El-Din & Farag, 2023*). Using a homogenizer (Ultra-Turrax, IKA, Staufen, Germany) at 11,000 rpm, 5 X 60 s with 1 min break intervals, about 150 g of the crushed sample was mixed with six mL methanol containing 10 µg/mL umbelliferone (Sigma-Aldrich, St. Louis, MO, USA, purity ≥ 98%) that used as an internal standard and for MS calibration. Further processing, the extract was centrifuged at $3,000 \times$ g for 30 min after being vortexed for 1 min, filtered through a 22 µm pore size filter, and directly used for HR-UPLC-MS/MS analysis. For GC/MS, 100 µl was aliquoted in a glass vial and left to evaporate till dryness under a nitrogen stream. For bioassay, fruit pulp was extracted using 100% MeOH (Sigma Aldrich, St. Louis, MO, USA) till exhaustion and evaporated using rotavap under vacuum to yield dried yellowish residue stored at −20 °C till further assays.

### Chemicals and fibers

The stableflex fiber used for solid phase micro-extraction (SPME) was covered by divinylbenzene/carboxen/polydimethylsiloxane (DVB/CAR/PDMS, 50/30 µm) and was obtained from Supelco (Oakville, ON, Canada). Chemicals were acquired from Sigma Aldrich (St. Louis, MO, USA). Milli-Q water and solvents were used for HR-UPLC/MS/MS analysis; formic acid and acetonitrile were of LC-MS grade and obtained from J. T. Baker (The Netherlands).

ABTS [2,20′azino-bis (3-ethylbenzothiazoline-6-sulfonic acid) diammonium salt] ≥ 98% purity, DPPH (2,2-diphenyl-1-picrylhydrazyl), ferric chloride for FRAP (ferric reducing antioxidant power), Trolox (6-hydroxy-2,5,7,8-tetramethyl-chromane-2-carboxylic acid; ≥97% purity), porcine pancreatic lipase (PL) enzyme type II, intestinal $\alpha$-glucosidase, Orlistat, Acarbose from Sigma Aldrich Chemie GmbH (St. Louis, MO, USA).

### GC/MS analysis of silylated primary polar metabolites of *M. zapota* fruit pulp

Analysis of primary metabolites in fruit pulp followed the exact procedure (*El-Akad, El-Din & Farag, 2023*), in triplicates under the same conditions. The derivatization process was compiled as follows; the dried methanol extract of fruits was derivatized using a silylating agent; N-methyl-N-(trimethylsilyl)-trifluoroacetamide (MSTFA) (150 µL equally diluted with anhydrous pyridine), incubated in an oven for 45 min at 60 °C (Yamato Scientific

DGS400 Oven, QTE TECHNOLOGIES, Hanoi, Vietnam), just before GC/MS analysis. Silylated compounds were separated on a column 30 m × 0.25-mm id × 0.25-m film (Rtx-5MS Restek, Bellefonte, PA, USA), and were analyzed under conditions described in *Farag et al. (2022)*. Analysis was done in triplicate under the same conditions along with a blank sample to evaluate the variation in biological samples.

## SPME-GC/MS analysis of volatiles in *M. zapota* fruit pulp

Preparation and analysis of the aroma profile in fruit pulp were performed following the same conditions reported in (*Farag et al., 2022*). A quadrupole mass spectrometer connected to an Agilent 5977B GC/MSD (Santa Clara, CA, USA) was used fitted with a DB-5 column (Supelco, Bellefonte, PA, USA) 30 m × 0.25 mm i.d. × 0.25 m film thickness. The scan range of the MS spectrometer was adjusted at *m/z* 40–500 and EI mode at 70 eV. Analysis was done in triplicate under the same conditions along with a blank sample to evaluate the variation in biological samples.

## Identification of volatile and non-volatile silylated components using GC/MS

Deconvolution of the GC/MS spectrum was first applied using AMDIS software (http://www.amdis.net). Detection of compounds was achieved by matching the retention indices (RI) of the detected peaks with those of the n-alkanes series (C8-C30), along with matching their mass spectra with respected databases; NIST011 and WILEY libraries, and standards whenever available.

## High-resolution ultra high-performance liquid chromatography coupled with tandem mass spectrometry (HR-UPLC/MS/MS) analysis of non-polar metabolites

HR-UPLC/MS/MS analysis was performed using an ACQUITY UPLC system (Waters, Milford, MA, USA) coupled with an HSS T3 ($C_{18}$) reversed-phase sorbent column (100 × 1.0 mm, particle size 1.8 $\mu$m; Waters). The analysis was accomplished following the precise guidelines as reported by *Hegazi et al. (2022)*. The tentative identification of compounds was based on the generated molecular formula at an error of 10 ppm or less, and by comparing $MS^2$ fragments with reported literature (*Böcker & Dührkop, 2016*).

## Molecular networking and metabolites' annotation of HR-UPLC/MS/MS data

The HR-UPLC/MS/MS data (acquired in positive ion mode) from the fruit extract was used to create a molecular network (MN) using the GNPS website (http://gnps.ucsd.edu). The raw data underwent conversion to an open-source format (.mzML) using the MS Convert package (Proteowizard Software Foundation, Version 3.0.19330, USA). The transformed (.mzML) files were then uploaded to the GNPS platform using WinSCP (SFTP, FTP, WebDAV, and SCP client). GNPS parameters included fragment ion tolerance (0.5 u), minimum-matched fragments (four ions), minimum pairs cosine score (0.65), and parent mass tolerance (1.0 u), which were used to generate consensus spectra.

The spectral network was visualized with the aid of Cytoscape 3.9.1. Each spectrum was represented as a node in the visualization, with spectrum-to-spectrum connections forming

edges based on structural correspondence identified through MS analysis (*Xu et al., 2021*; *Zia-ur Rehman Gurgul et al., 2022*). For natural products dereplication, various databases were searched, including PubChem (https://pubchem.ncbi.nlm.nih.gov/), Metabolome database (https://hmdb.ca/), online lipid calculator database (http://www.mslipidomics.info/lipid-calc/), and LIPID MAPS (https://www.lipidmaps.org/).

For visualization of metabolite classes with sapota fruit, acquired tandem mass spectrometry data, and molecular networks were (MNS) constructed. In MNS, the mass spectrometric data were clustered according to the resemblances of their fragmented ions (*Ragheb et al., 2023*), and to aid in the identification of unknown peaks.

## Total phenolic and total flavonoid contents assay

Fruit pulp extract was analyzed for total phenolic (TP) content and total flavonoid (TF) content, after being re-dissolved in 5% DMSO with 70% ethanol to yield a concentration of 1 mg/mL stock solution.

The evaluation of TP content was based on the Folin-Ciocalteu method, previously described by *Babotă et al. (2018)*. 20 µL sample were added to 100 µL Folin–Ciocalteu reagent at a ratio of 1:9 V/V and mixed well. 80 µL of $Na_2CO_3$ solution (1%) was added to the previous mixture. After 30 min of incubation at room temperature, the absorbance was measured at 760 nm. The result was represented as milligrams of gallic acid equivalent per gram sample (mg GAE/g extract), after triple measurements (mean $\pm$ SD).

For TF content, aluminum chloride assay was used (*Babotă et al., 2018*). A 100 µL sample was added to 100 µL of 2% aluminum trichloride/methanol. After 10 min incubation at room temperature, the absorbance was measured at 415 nm. Results were expressed as milligrams of rutin equivalent per gram sample (mg RE/g extract), after triple measurements (mean $\pm$ SD). In the two tests, the absorbances were analyzed in 96-well plates (SPECTROstar® Nano Multi-Detection Microplate Reader; BMG Labtech, Ortenberg, Germany).

### *In vitro* antioxidant assays

Two assays depending on the free radical scavenging actions; DPPH (1,1-diphenyl-2-picrylhydrazyl) and ABTS (2,2′-azino-bis(3-ethylbenzothiazoline) 6-sulfonic acid), along with the ferric reducing antioxidant power (FRAP) technique for ferric reducing capacity, were applied following the protocols of *Babotă et al. (2018)*. Initially, the extract was dissolved in 70% ethanol to get a concentration of 1mg/mL. The absorbances were measured in 96-well plates (SPECTROstar® Nano Multi-Detection Microplate Reader; BMG Labtech, Ortenberg, Germany). A calibration curve was established using different concentrations of Trolox, a standard antioxidant substance. The resulting data were reported as mg of Trolox equivalents per gram sample (mg TE/g extract) in each case, after triple measurements (mean $\pm$ SD).

The DPPH free radical scavenging assay: 30 µL of the fruit methanol extract was mixed with 0.004% DPPH methanol solution and incubated for 30 min in the dark at room temperature. Absorbance was measured at 517 nm.

For ABTS free radical scavenging assay: $ABTS^+$ radical was prepared by adding 2.45 mM potassium persulfate to 7 mM ABTS solution, leaving the mixture to stand in the dark

and room temperature for 12 h. The resulting solution was further diluted with distilled water till absorbance reached $0.7 \pm 0.02$ at 734 nm and then mixed with fruit extract and incubated for 30 min at room temperature. The absorbance was measured at 734 nm.

For FRAP reducing capacity: FRAP reagent was prepared by blending acetate buffer (0.3 M, pH 3.6), 2,4,6-tris(2-pyridyl)-s-triazine (TPTZ) (10 mM) mixed with ferric chloride (20 mM) in 40 mM HCl at a ratio of 10:1:1(V/V/V). The reagent was added to the diluted fruit sample, mixed, and incubated for 30 min of at room temperature and absorbance was measured at 593 nm.

### *In vitro* enzyme inhibition assays

The crude porcine PL type II enzyme was suspended in Tris-HCL buffer (2.5 mmol, adjusting pH at 7.4 with 2.5 mmol NaCl), mixing with a stirrer for 15 min, to get a concentration of 200 U/mL (5 mg/mL). The sample was incubated with 0.1 mL of PL solution for 5 min at 37 °C then *p*-nitrophenyl butyrate (*p* NPB) substrate (10 mM in acetonitrile) was added. Inhibition activity was measured colorimetrically based on *p*-nitrophenol release (at 410 nm compared to a blank of denatured enzyme), following a modified method mentioned by *Bustanji et al. (2011)*. The experiment was run in triplicate and percentage inhibition represented the average of three observations using two concentrations at 5 & 10 mg/mL of the extract, expressed in terms of $IC_{50}$ (Half-maximal inhibitory concentration). Orlistat was used as a positive control as a standard PL inhibitor.

$\alpha$-glucosidase inhibitory action was measured following the previously reported protocol (*Tanase et al., 2019*). In a 96-well plate, 50 µL of yeast $\alpha$-glucosidase (1 U/mL) was mixed with equal volumes of diluted sample, 100 mM phosphate buffer (pH 6.8), and the substrate (5 mM *p*-nitrophenol-$\alpha$-D-glucopyranoside (*p*-NPG). After an incubation period at 37 °C for 20 min, the color was formed due to *p*-nitrophenol release at 405 nm, and acarbose was used as a positive control. $IC_{50}$ value was calculated from three measurements of the two tested extract doses at 5 & 10 mg/mL. In both tests, the results were calculated as the concentration that inhibited 50% of the enzyme according to the equation. Percentage of enzyme inhibition = (Ac -As / Ac) X 100, where Ac is the absorbance of the control and As is the absorbance of the sample.

## RESULTS & DISCUSSION

### Metabolites profiling of silylated primary polar metabolites in *M. zapota* fruit pulp as analyzed *via* GC/MS

GC/MS analysis of the non-volatile primary metabolites in fruit was carried out post-silylation to present a comprehensive overview of metabolites (Fig. 1), and to further account for nutritional and sensory attributes in fruits. As listed in Table 1, 68 compounds belonging to 11 chemical classes were detected. The most abundant metabolite classes included sugars, sugar acids, and sugar alcohols detected at 74.0, 18.27, and 7%, respectively. Other detected primary metabolites though at much lower levels included fatty acids/esters

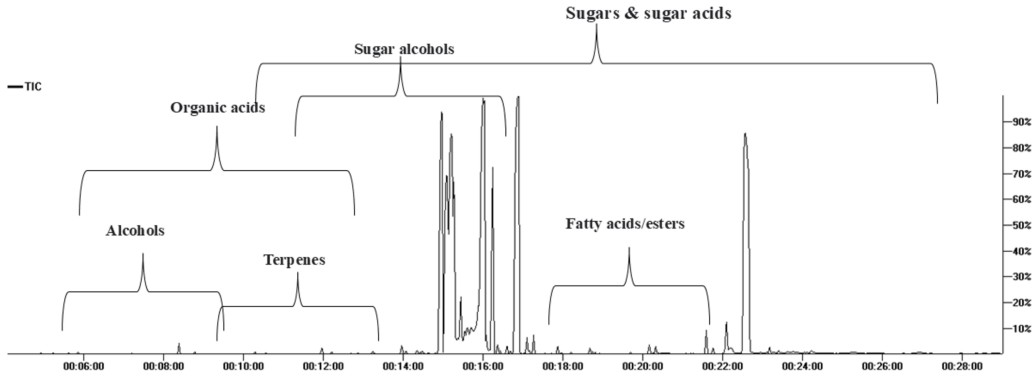

**Figure 1** **Total ion chromatogram (TIC) of *M. zapota* fruit silylated polar metabolites analyzed using GC/MS.**

(0.22%), organic acids (0.22%), and inorganic acids (0.124%), along with traces of alcohols, terpenes, and nitrogenous compounds.

The high sugars' level in fruit imparts a sweet taste as typical in most fruits. As represented in the TIC (Fig. 1), sugars represented major primary metabolites detected in 22 peaks, especially mono-sugars to account for 75% of identified sugars. The most prominent forms included fructose (22.1%), D-glucose (16.6%), and mannose (16.5%). Sucrose (17.9%, peak 63) was the predominant disaccharide. Previous reports revealed that the total sugars in sapota fruit comprised *ca*. 46 to 52.2% of its weight (*Jadhav, Swami & Pujari, 2018*).

The high level of sugar acids represented by keto-D-gluconic acid (9.8%) and L-gluconic acid lactone (7.9%) imparts a slightly tangy and acidic taste, which might provide a balanced sensation alongside the intense sweet taste (*Karaffa & Kubicek, 2021*). Sugar alcohols with lower calorie intake than free sugars were represented by 5-deoxy-myo-inositol (6.7%). In addition to its low-calorie level, it reduces the body's resistance to insulin and aids in diabetes management (*Corrado & Santamaria, 2015*). Although organic acids were present at minor levels (0.22%), they were represented by 13 compounds, with oxalic and pyruvic acids as major forms.

Fatty acids/esters composition plays a role in nutrition and flavor in fruits, detected at 0.23% including glycerol monostearate, the monoglyceride ester with a sweet taste. Likewise, saturated fatty acid palmitic acid (0.045%) and its monoester, monopalmitin (0.054%), were detected suggesting that fruit is enriched with saturated fatty acids. To the best of our knowledge, this is the first detailed report on primary metabolites composition in sapota fruit.

## Aroma profiling of *M. zapota* fruits pulp *via* SPME coupled to GC/MS

SPME-GC/MS analysis of aroma composition in *M. zapota* fruit revealed the detection of 17 compounds belonging to 8 chemical classes mostly dominated by nitrogenous compounds

**Table 1  Identified silylated polar metabolites in *M. zapota* fruits using GC/MS, results expressed as a relative percentile % of the total peak area (*n* = 3).** Results are represented as a relative percentile of the whole peak area (*n* = 3).

| Peak | Rt. (min.) | KI | Metabolite | Average ± SD |
|------|-----------|-----|------------|--------------|
|  |  |  | **Alcohols** |  |
| 1 | 4.81 | 1,042 | 2,3-Butanediol, 2TMS | tr. |
| 2 | 4.93 | 1,049 | 2,3-Butanediol, 2TMS isomer | 0.036 ± 0.002 |
| 3 | 5.127 | 1,061 | 1,3-Propanediol, 2TMS | tr. |
| 10 | 7.201 | 1,190 | 1,2-Glycerol, 2TMS | 0.014 ± 0.001 |
| 17 | 8.654 | 1,293 | Butanetriol, 3TMS | 0.009 ± 0.001 |
|  |  |  | **Total** | 0.06 ± 0.005 |
|  |  |  | **Fatty acids/esters** |  |
| 52 | 17.14 | 2,030 | Palmitic acid, TMS | 0.045 ± 0.003 |
| 55 | 18.637 | 2,192 | Linoleic acid, TMS | tr. |
| 62 | 21.751 | 2,573 | 1-Monopalmitin, 2TMS | 0.054 ± 0.020 |
| 64 | 23.164 | 2,765 | Glycerol monostearate, 2TMS | 0.121 ± 0.057 |
|  |  |  | **Total** | 0.225 ± 0.081 |
|  |  |  | **Organic acids** |  |
| 4 | 5.238 | 1,068 | Lactic Acid, 2TMS | 0.027 ± 0.004 |
| 5 | 5.45 | 1,081 | Glycolic acid, 2TMS | tr. |
| 6 | 5.63 | 1,092 | Pyruvic acid, 2TMS | 0.048 ± 0.003 |
| 7 | 5.86 | 1,107 | Oxalic acid, 2TMS | 0.071 ± 0.020 |
| 8 | 6.26 | 1,131 | Methylmalonic acid, TMS | tr. |
| 9 | 6.693 | 1,159 | Hydroxybutyric acid, 2TMS | 0.006 ± 0.001 |
| 13 | 7.921 | 1,240.5 | Benzoic acid, 3TMS | 0.006 ± 0.001 |
| 18 | 8.785 | 1,302.5 | Malonic acid, 3TMS | 0.020 ± 0.005 |
| 19 | 8.859 | 1,308 | Succinic acid, 2TMS | 0.004 ± 0.002 |
| 22 | 10.185 | 1,403 | Ketosuccinic acid, TMS | 0.002 ± 0.000 |
| 24 | 11.24 | 1,488 | Malic acid, 3TMS | tr. |
| 26 | 12.207 | 1,566 | Erythronic acid, 4TMS | 0.025 ± 0.004 |
| 27 | 12.697 | 1,606 | Tartaric acid, 4TMS | tr. |
|  |  |  | **Total** | 0.220 ± 0.040 |
|  |  |  | **Inorganic acid** |  |
| 15 | 8.39 | 1,274 | Phosphoric acid, tri-TMS | 0.124 ± 0.014 |
|  |  |  | **Nitrogenous compounds/ Amino acid** |  |
| 11 | 7.589 | 1,216.7 | Uracil, TMS | 0.002 ± 0.0 |
| 12 | 7.78 | 1,230 | Urea, 2TMS | 0.002 ± 0.0 |
| 14 | 8.102 | 1,253 | L-Serine, 2TMS | 0.001 ± 0.0 |
|  |  |  | **Total** | 0.005 ± 0.001 |
|  |  |  | **Terpenes** |  |
| 16 | 8.517 | 1,283 | Anethole | 0.001 ± 0.001 |
| 21 | 9.3 | 1,339.4 | α-Terpinyl acetate | 0.002 ± 0.0 |
| 32 | 13.712 | 1,681 | Curlone | 0.002 ± 0.003 |
|  |  |  | **Total** | 0.005 ± 0.005 |

**Table 1** (*continued*)

| Peak | Rt. (min.) | KI | Metabolite | Average ± SD |
|------|-----------|-----|-----------|--------------|
| | | | **Sugar acids** | |
| 20 | 9.18 | 1,331 | Glyceric acid, 3TMS | 0.005 ± 0.0 |
| 39 | 14.604 | 1,778 | 2-Keto-l-gluconic acid, 5TMS | 0.008 ± 0.0 |
| 43 | 15.167 | 1,831.3 | L-gluconic acid, 4TMS, lactone | 7.934 ± 0.204 |
| 41 | 15.069 | 1,821.6 | Mannonic acid, 5TMS, lactone | 0.008 ± 0.007 |
| 45 | 15.182 | 1,833 | Keto-gluconic acid, 5TMS | 9.790 ± 1.663 |
| 51 | 17.087 | 2,024 | D-Gluconic acid, 6TMS | 0.101 ± 0.007 |
| 53 | 17.25 | 2,041.6 | D-Glucuronic acid, 4TMS | 0.292 ± 0.019 |
| 56 | 18.73 | 2,202 | D-Galacturonic acid, 5TMS | 0.041 ± 0.005 |
| 60 | 20.315 | 2,390 | D-Glucuronic acid, 5-TMS | 0.095 ± 0.021 |
| | | | **Total** | 18.274 ± 1.925 |
| | | | **Sugar alcohols** | |
| 25 | 11.96 | 1,546.5 | Deoxyribitol, 4TMS | 0.026 ± 0.003 |
| 34 | 13.963 | 1,720 | Arabitol, 5TMS | 0.128 ± 0.018 |
| 35 | 14.061 | 1,728.9 | D-Glucitol, 6-deoxy, 5TMS | 0.048 ± 0.004 |
| 36 | 14.343 | 1,754.3 | D-Mannitol, 6TMS | 0.010 ± 0.001 |
| 40 | 14.95 | 1,809.6 | Myo-inositol, 5-deoxy, 5TMS | 6.691 ± 0.292 |
| 54 | 17.859 | 2,108 | Myo-Inositol, 6TMS | 0.096 ± 0.011 |
| | | | **Total** | 7.000 ± 0.329 |
| | | | **Sugars** | |
| 23 | 10.227 | 1,407 | L-Threose, 3TMS | 0.043 ± 0.003 |
| 28 | 12.87 | 1,622 | Arabinose, 4TMS | 0.020 ± 0.002 |
| 29 | 13.234 | 1654.6 | Arabinopyranose, 4TMS | 0.049 ± 0.007 |
| 30 | 13.277 | 1,658 | Galactopyranose, 5TMS | 0.002 ± 0.0 |
| 31 | 13.449 | 1,674 | Arabinofuranose, 4TMS | 0.003 ± 0.0 |
| 33 | 13.774 | 1,695 | L-Rhamnose, 4TMS | 0.002 ± 0.0 |
| 37 | 14.473 | 1,766 | 1-Deoxyglucose, 4TMS | 0.012 ± 0.002 |
| 38 | 14.52 | 1,770 | Mannopyranose, 6-deoxy, 4TMS | 0.007 ± 0.0 |
| 42 | 15.077 | 1,822.4 | Fructofuranose, 5TMS isomer | 0.003 ± 0.003 |
| 44 | 15.172 | 1,831.6 | Fructofuranose, 5TMS | 11.392 ± 0.212 |
| 46 | 15.245 | 1,839 | Fructofuranose, 5TMS isomer | 5.919 ± 0.248 |
| 47 | 15.429 | 1,857 | D- Galactofuranose, 5TMS | 0.207 ± 0.023 |
| 48 | 16.064 | 1,920 | Mannose, 5TMS | 16.530 ± 0.091 |
| 49 | 16.225 | 1,936 | D-Fructose, 5TMS | 4.791 ± 0.214 |
| 50 | 16.87 | 2,000.5 | D-Glucose, 5TMS | 16.652 ± 0.457 |
| 57 | 18.8 | 2,211 | Cellobiose, 8TMS | 0.006 ± 0.003 |
| 61 | 21.58 | 2,551 | Turanose, 8TMS | 0.254 ± 0.057 |
| 63 | 22.559 | 2,682 | Sucrose, 8TMS | 17.963 ± 1.072 |
| 65 | 23.393 | 2,797 | 3-α-Mannobiose, 8TMS | 0.033 ± 0.001 |
| 66 | 24.207 | 2,908.7 | Melibiose, 8TMS | 0.057 ± 0.004 |
| 67 | 28.528 | 3,503 | Maltose, 8TMS | 0.006 ± 0.003 |
| 68 | 28.7 | 3,527 | β-Gentiobiose, 8TMS | 0.052 ± 0.046 |
| | | | **Total** | 74.002 ± 2.449 |

| Peak | Rt. (min.) | KI | Metabolite | Average ± SD |
|------|-----------|------|-----------|--------------|
|  |  |  | **Glycerolipids** |  |
| 58 | 19.685 | 2,315 | Glycerol- α galactopyranoside, 6TMS | 0.041 ± 0.004 |
| 59 | 20.155 | 2,371 | Glycerol-galactopyranoside isomer, 6-TMS | 0.044 ± 0.004 |
|  |  |  | **Total** | 0.084 ± 0.008 |

**Notes.**
Tr, Traces

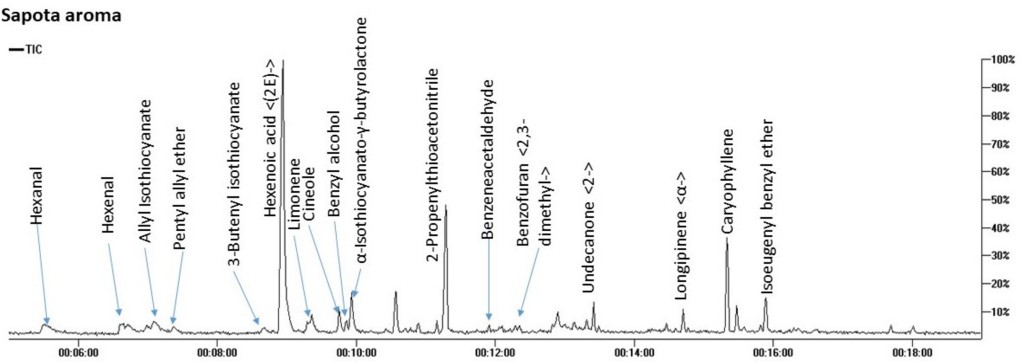

**Figure 2** **Total ion chromatogram (TIC) of *M. zapota* fruit volatile constituents analyzed using SPME-GC/MS.**

amounting to 71.7%. Other classes included ethers (7.8%), terpenes (7.6%), and aldehydes (5.8%) as represented in Fig. 2 and Table 2.

The identified nitrogenous compounds (peaks 3, 5, 10, and 11) were detected for the first time in the fruit belonging to isothiocyanates, a hydrolysis product of glucosinolates. The major compound was 3-butenyl isothiocyanate (64.3%), alongside allyl isothiocyanate (2.8%). The presence of ethers in fruit provides specific fragrances, represented by benzyl isoeugenol ether (4.2%), peak 17, in addition to pentyl allyl ether (3.1%), peak 4, and cineole, peak 8 (*Kirsch & Buettner, 2013*).

As typical in fruit aroma, a considerable amount of mono- and sesquiterpenes were detected amounting (7.6%) of the total aroma composition, with β-caryophyllene (5.42%), peak 16, and limonene (1.54%) peak 7 as major components. β-Caryophyllene was previously detected in sapota fruit volatiles using steam distillation (*Pino, Marbot & Aguero, 2003*). Aldehydes, which accounted for 5.8% of the total fruit aroma, likely contribute to the fruit scent and likewise, protect against their deterioration due to potential antibacterial action (*Aljaafari et al., 2022*). The major form was hexanal at 4% to impart an apple-like odor (*Plotto, Bai & Baldwin, 2017*), alongside benzyl alcohol (2.6%) of a light fragrant smell (*Kulkarni & Mehendale, 2005*) and all to contribute to sapota fruit-specific scent.

**Table 2  Volatile compounds in *M. zapota* fruits as analyzed by SPME coupled to GC/MS.** Results are represented as a relative percentile of the whole peak area ($n = 3$).

| Peak | Rt. (min.) | KI | Metabolite | Percent ±SD |
|------|-----------|-----|-----------|-------------|
| | | | **Aldehydes** | |
| 1 | 5.492 | 913 | Hexanal | 4.133 ± 0.96 |
| 2 | 6.603 | 1,096 | 3-Hexenal, (Z)- | 1.208 ± 0.52 |
| 12 | 12.29 | 1,573 | Benzene acetaldehyde | 0.438 ± 0.15 |
| | | | Total | 5.78 ± 1.62 |
| | | | **Nitrogenous compounds** | |
| 3 | 7.083 | 1,176 | Allyl Isothiocyanate | 2.80 ± 0.22 |
| 5 | 8.94 | 1,313 | 3-Butenyl isothiocyanate | 64.33 ± 1.96 |
| 10 | 10.56 | 1,433 | α-Isothiocyanato-γ-butyrolactone | 4.54 ± 0.35 |
| 11 | 11.29 | 1,492 | 2-Propenylthioacetonitrile | 0.02 ± 0.01 |
| | | | Total | 71.69 ± 2.55 |
| | | | **Ethers** | |
| 4 | 7.136 | 1,186 | Pentyl allyl ether | 3.14 ± 0.29 |
| 8 | 9.86 | 1,379 | Cineole <1,8-> | 0.46 ± 0.12 |
| 17 | 15.88 | 1,902 | Isoeugenyl benzyl ether | 4.23 ± 0.79 |
| | | | Total | 7.825 ± 1.20 |
| | | | **Acids** | |
| 6 | 9.35 | 1,343 | Hexenoic acid <(2E)-> | 1.73 ± 0.51 |
| | | | **Terpenes** | |
| 7 | 9.75 | 1,371 | Limonene | 1.54 ± 0.27 |
| 15 | 14.699 | 1,786 | Longipinene <α-> | 0.62 ± 0.09 |
| 16 | 15.33 | 1,847 | Caryophyllene | 5.42 ± 0.63 |
| | | | Total | 7.59 ± 0.99 |
| | | | **Alcohol** | |
| 9 | 9.93 | 1,384 | Benzyl alcohol | 2.59 ± 0.41 |
| | | | **Furan** | |
| 13 | 12.89 | 1,623 | Benzofuran <2,3-dimethyl-> | 2.01 ± 0.56 |
| | | | **Ketone** | |
| 14 | 13.48 | 1,677 | Undecanone <2-> | 0.79 ± 0.34 |

## Non-polar metabolites profiling of *M. zapota* fruit as analyzed *via* HR-UPLC/MS/MS

Considering that GC/MS can only detect low molecular weight polar phytochemicals in food, and to provide a comprehensive composition of sapota fruit metabolome, HR-UPLC/MS/MS was employed to complement GC/MS and target large molecular weight lipids (*Islam et al., 2021*). Herein, a list of tentatively identified metabolites of *M. zapota* fruit is presented in Table 3, along with their chromatographic and spectroscopic data (Fig. 3). Major identified metabolites contained lipoidal components *e.g.*, fatty acyl amides, phospholipids, and sphingolipids, and contrary to low levels of lipids detected using GC/MS more suited for polar chemicals profiling. Other classes detected at minor levels included fatty acyl esters, nitrogenous compounds, glycol, amino acids, and diethanolamines. To aid in metabolites assignment, molecular networking was used for HR-UPLC/MS/MS

Farag et al. (2024), *PeerJ*, DOI 10.7717/peerj.17914

Peerj

**Table 3** **Major non polar metabolites annotated in *M. zapota* fruit methanol extract via HRUPLC/MS/MS in positive ion mode.**

| Peak No. | Rt (min.) | Mol. Ion (M+H)$^+$ | Error (ppm) | Molecular formula | MS/MS fragments | Identification | Class |
|---|---|---|---|---|---|---|---|
| 1 | 0.987 | 166.0862 | +0.33 | $C_9H_{11}NO$ | 121, 120, 103 | Phenylalanine | Amino acid |
| 2 | 2.476 | 139.0755 | −1.04 | $C_8H_{10}O_2$ | 124, 121, 120 | Styrene glycol | Alcohol |
| 3 | 4.622 | 262.2371 | +2.18 | $C_{14}H_{31}NO_3$ | 226, 122 | Tetradecaphytosphingosine | Sphingolipid |
| 4 | 5.459 | 230.2474 | +1.92 | $C_{14}H_{31}NO$ | 213, 212, 109 | Unknown | Nitrogenous |
| 5 | 5.631 | 290.2688 | +0.59 | $C_{16}H_{35}NO_3$ | 272, 254,242 | *LCBs (16;0)* | Sphingolipid |
| 6 | 5.645 | 272.2576 | +2.97 | $C_{16}H_{33}NO_2$ | 254, 236, 224 | Hexadecasphingosine | Sphingolipid |
| 7 | 5.73 | 288.2527 | +2.16 | $C_{16}H_{33}NO_3$ | 227, 116, 102 | Unknown amide | Nitrogenous lipid |
| 8 | 5.964 | 316.2836 | +3.24 | $C_{18}H_{37}NO_3$ | 298,286, 281, 280, 262, 256, 141 | Dehydrophytosphingosine | Sphingolipid |
| 9 | 6.388 | 302.3047 | +2.18 | $C_{18}H_{39}NO_2$ | 284, 106, 102 | Tetradecyldiethanolamine | Nitrogenous lipid |
| 10 | 6.41 | 318.2997 | +1.8 | $C_{18}H_{39}NO_3$ | 300, 282, 264 | Phytosphingosine | Sphingolipid |
| 11 | 6.624 | 300.2891 | +2.02 | $C_{18}H_{37}NO_2$ | 282, 264 | Sphingosine | Sphingolipid |
| 12 | 7.063 | 415.2108 | +2.48 | $C_{20}H_{33}NO_6P$ | 354 | Unknown PE | Phospholipid |
| 13 | 7.145 | 330.3361 | +1.69 | $C_{20}H_{43}NO_2$ | 312,106,102 | Hexadecyl diethanolamine | Nitrogenous lipid |
| 14 | 7.289 | 478.2936 | −1.64 | $C_{23}H_{45}NO_7P$ | 337 | LysoPE(0:0/18:2) | Phospholipid |
| 15 | 7.833 | 454.2918 | +2.24 | $C_{21}H_{44}NO_7P$ | 313 | LysoPE(0:0/16:0) | Phospholipid |
| 16 | 8.014 | 358.368 | −0.12 | $C_{22}H_{47}NO_2$ | 340, 322, 270 | Unknown | Unknown |
| 17 | 10.59 | 256.2632 | +1.14 | $C_{16}H_{33}NO$ | 239, 238,209, 116, 102 | Palmitamide | Fatty acyl amide |
| 18 | 10.90 | 282.2785 | +2.28 | $C_{18}H_{35}NO$ | 247, 135, 121, 111, 102 | Octadecenamide (Oleamide) | Fatty acyl amide |
| 19 | 10.94 | 331.2843 | −0.04 | $C_{19}H_{38}O_4$ | 313, 109 | Hexadecanoyl- glycerol | Fatty acyl ester |
| 20 | 11.10 | 512.503 | +1.41 | $C_{32}H_{65}NO_3$ | 284 | Tetradecanoyl-sphinganine | Sphingolipid |
| 21 | 11.90 | 284.2937 | +3.85 | $C_{18}H_{37}NO$ | 200, 174,130, 116, 102 | Octadecanamide (Steramide) | Fatty acyl amide |
| 22 | 12.02 | 310.3099 | +1.75 | $C_{20}H_{39}NO$ | 293,292, 275, 268, 247, 135,121,111,109 | Eicosenamide | Fatty acyl amide |
| 23 | 12.25 | 359.3153 | +0.8 | $C_{21}H_{42}O_4$ | 341, 267, 239,112, 109 | Glyceryl Monostearate | Fatty acyl ester |
| 24 | 12.36 | 540.5345 | +0.97 | $C_{34}H_{69}NO_3$ | 307,286,285,284 | Palmitoylsphinganine | Sphingolipid |
| 25 | 12.40 | 568.566 | +0.57 | $C_{36}H_{73}NO_3$ | 285, 284, 264 | Heneicosanoyl-pentadecasphinganine | Sphingolipid |
| 26 | 13.08 | 312.3258 | +0.94 | $C_{20}H_{41}NO$ | 182, 116,112, 102 | Icosanamide | Fatty acyl amide |
| 27 | 13.15 | 338.3414 | +1.01 | $C22H43NO$ | 339, 321,320,303. 265,247, 135, 121 | Erucamide | Fatty acyl amide |
| 28 | 13.16 | 675.6761 | +0.16 | $C_{44}H_{86}N_2O_2$ | 338, 321, 303,121, 111,109 | Erucamide dimer | Fatty acyl amide |
| 29 | 13.27 | 732.5612 | +1.13 | $C_{40}H_{77}NO_{10}$ | 570, 552, 314, 262 | Cerebroside(araliacerebroside) | Sphingolipid |
| 30 | 14.14 | 302.305 | +1.18 | $C_{18}H_{39}NO_2$ | 285,284, 217 | Octadecasphinganine | Sphingolipid |
| 31 | 14.34 | 429.3715 | +2.82 | $C_{29}H_{48}O_2$ | 401, 371, 345, 205, 203, 187,165 | Dehydrotocopherol | Tocopherol |

**Notes.**

LCBs, long-chain bases sphingolipid

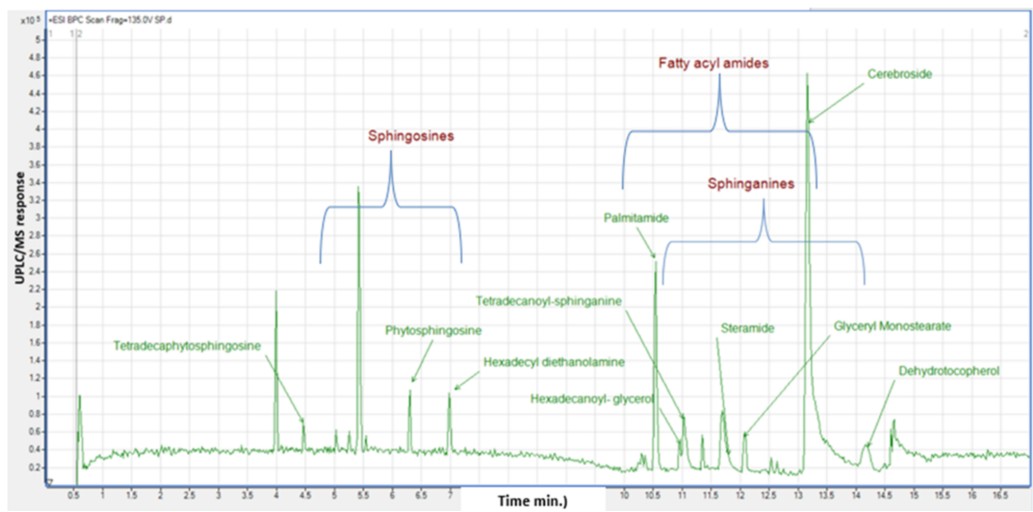

**Figure 3** Base peak chromatogram (BPC) of *M. zapota* fruit non polar metabolites analyzed using HR UPLC/MS/MS, in positive ion mode.

dataset visualization. The MN afforded a total of 346 nodes, of which 141 clustered nodes and 205 self-looped nodes were detected (Fig. 4). The visual aid of MNS showed the diverse metabolite classes, which assisted in analog identification. The substantial clusters of positive MN belonged to oxylipids including cluster A (sphingosine and sphinganine), cluster B (fatty acyl amides), cluster C (phytosphingosine), and cluster D (fatty acyl esters), (Fig. 4).

*Identification of fatty acyl amides*

Fatty acyl amides, a subclass of lipids, exist as bioregulators for lipids in plants and are formed through amidation of fatty acids (*Tanvir, Javeed & Rehman, 2018*). Seven fatty acyl amides were identified in sapota fruit extract based on neutral losses of 14 amu, indicative of an acyl group (Fig. S1). Further, the annotation of saturated fatty acyl amides [M + H]$^+$ at *m/z* 256.263, 284.293, and 312.325 was based on their abundant fragments at *m/z* 102 ($C_5H_{10}NO$) and *m/z* 116 ($C_6H_{12}NO$). The presence of a single unsaturation in the alkyl chain of acyl amides alters product ion dramatically, as fragmentation differed with daughter ions corresponding to the combined neutral losses (−35 Da) of $H_2O$ and $NH_3$ in the amide group. Distinct fragments at *m/z* 247 for the successive losses of water and ammonia moieties along with multiple losses of $CH_2$ were recognized in MS$^2$ spectra of assigned unsaturated acyl amides. Conclusively, the whole loss of the acyl chain and formation of 9-carbon and 10-carbon macrocyclic dieneyl cation yielded daughter ions at *m/z* 135 and 121 and aided in structural elucidation of that subclass (*Murphy, 2014*).

Peaks 17, 18, 21, 22, 26, 27 and 28 exhibited molecular ions [M+H]$^+$ at *m/z* 256.26, 282.27, 284.29, 310.30, 312.325, 338.34 and 675.67 in MS/MS spectra with distinctive fragment ions of fatty acyl amides; palmitamide, octadecenamide (oleamide), octadecanamide (steramide), eicosenamide icosanamide, erucamide and erucamide dimer, respectively, cluster B in MN (Fig. 4). These metabolites are reported here for the first time

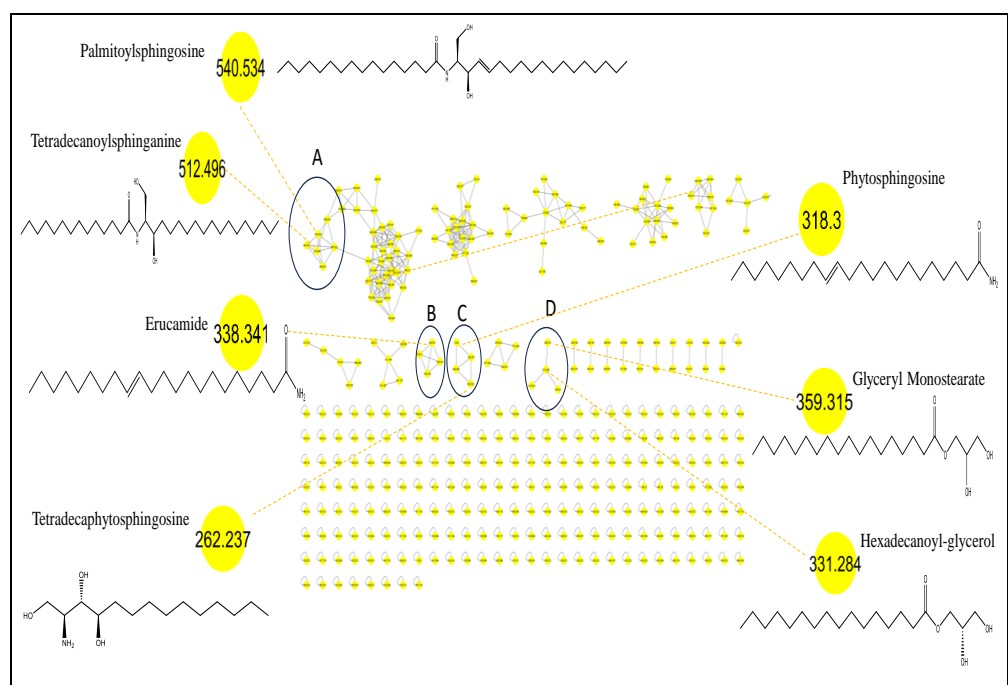

**Figure 4  Molecular networks created using MS/MS data from *M. zapota* fruit.**

in sapota fruit, and likely to account for a wide array of therapeutic indications such as treatment of bacterial infections, cancer, inflammations, and metabolic disorders (*Tanvir, Javeed & Rehman, 2018*). Steramide was detected previously in sapota leaves (*Tamsir et al., 2020*).

### Identification of sphingolipids

The identified sphingolipids were detected in clusters A and C in the GNPS network (Fig. 4). Sphingosine is the major form present in this class and is assigned in peaks (3, 6, 8, 10, and 11), followed by the sphinganine class which was observed in BPC in peaks (20, 24, 25, and 30). The lipophilicity, formula composition, and fragmentation pattern suggest that these peaks are sphingolipid conjugates.

Most of the sphingolipids and their dihydro equivalents fragment to backbone ions with $m/z$ 264 in positive ion mode as a key for the identification of sphingolipids (*Otify et al., 2019*). Most notably, product ion ($m/z$ 284) is for sphinganine, whereas product ion at $m/z$ 282 corresponds to sphingosine (Fig. S2).

For example, peak 8 exhibited a molecular formula [$C_{18}H_{37}NO_3$ ($m/z$ 316.2836)], such a formula matches the class of sphingoid bases (that is non-phosphorylated plant sphingolipids) belonging to basic sphingoid compounds, either dehydrophytosphingosine, or 6-hydroxysphingosine (*Lénárt et al., 2021*). The fragmentation pattern showed product ions at $m/z$ 280 and 262 corresponding to losses of 2 and 3 $H_2O$ molecules, respectively, and assigning it as dehydrophytosphingosine.

Peak 3 showed fragmentation pattern of tetradecaphytosphingosine, based on the neutral loss of two water molecules and an alkyl chain ($C_{10}H_{20}$, 140 Da) at $m/z$ 226 and 122 (Table 3). *Sphingolipid* long-chain base (LCB) was detected in peak 5 showing fragment ions at $m\backslash z$ 272, 254, 242 (*Qu et al., 2018*). The cerebroside (peak 29) with $(M+H)^+$ at $m\backslash z$ 732.56 ($C_{40}H_{77}NO_{10}$) and abundant ion at $m\backslash z$ 570 due to neutral loss of hexosyl and further loss of two water molecules to yield product ion at $m/z$ 534 (*Kang et al., 1999*) and assigned as araliacerebroside (Fig. S3).

C16 sphinganine, a sphingolipid conjugate, was identified previously in *M. zapota* leaves (*Tamsir et al., 2020*), this study represents the first comprehensive profiling of sphingolipids in sapota fruits.

### Identification of lysophosphatidylethanolamines

Lysophosphatidylethanolamines (Lyso-PE) were characterized in peaks 14 and 15 (Fig. S4) by the molecular formula of $C_xH_xNO_7P$ (*Ragheb et al., 2023*). LysoPE (0:0/18:2) and LysoPE (0:0/16:0) exhibited $(M+H)^+$ at $m/z$ 478.29 and 454.29, respectively. The most abundant ions at $m/z$ 337 and 313, in their positive-ion mass spectra, corresponded to the neutral loss of 141 Da of phosphoethanolamine (*Fang, Yu & Badger, 2003*), aiding in their assignment for the first time in sapota fruit.

### Identification of ethanolamines

Peak 9 with $[M + H]^+$ at $m/z$ 302 was assigned as tetradecyl diethanolamine ($C_{18}H_{39}NO_2$). The dehydration of the parent ion yielded $m/z$ 284, with further cleavage of the carbon chain to yield fragment ion at $m/z$ 102. The direct loss of carbon chain from quasi-molecular ion gave product ion at $m/z$ 106 (Fig. S5), a key fragment of this class (*Zhang et al., 2022*). Peak 13 showed a similar fragmentation pattern assigned as N-hexadecyl diethanolamine $[M + H]^+$ at $m/z$ 330.33 and fragment ions at $m/z$ 312, 106 and 102. This is the first report on the presence of ethanolamines in sapota fruit. Ethanolamines are at the hub of various cellular processes, they stimulate the synthesis of phosphoethanolamine, a vital component to maintain human health. Moreover, ethanolamine prevents cardiovascular disease and ischemia (*Patel & Witt, 2017*).

### Identification of fatty acyl esters

Fatty acyl esters were grouped in cluster D (Fig. 4) (peaks 23 & 19). This is the first report of the presence of fatty acyl esters in sapota fruit. Peak 23 showed the dehydration of precursor ion $[M + H]^+$ ($m/z$ 359) that yielded fragment ion at $m/z$ 341 (Fig. S6) attributed to an allylic cleavage and loss of glyceryl moiety yielding product ion at $m/z$ 267 assigned as glyceryl monostearate. Similarly, peak 19 displayed a similar fragmentation scheme, suggesting the presence of hexadecanoyl glycerol with product ions at $m/z$ 313 and 239.

### Identification of tocopherols

$MS^2$ fragments of dehydrotocopherol ($m/z$ 429.37) were detected in peak 31 and characterized by successive losses of alkyl groups to show fragmentation pattern; ($m/z$ 401, 345, and 303), eventually the complete loss of side-chain together with the cleavage of chromene ring developed the product ion $m/z$ 165 (Fig. S7).

## Total phenolic and total flavonoid contents

The quantitative estimation of total phenolics and flavonoids in sapota fruit flesh extract revealed that it encompasses a moderate amount of phenolics ($6.79 \pm 0.12$ mg GAE/g) and traces of flavonoids below our LOQ (limit of quantitation). The ripeness of the fruit results in a major change in its composition from an astringent taste owing to tannins and catechins, to a sweet taste due to the elevated sugar content. Fruit ripening had an impact on phenolic content due to the oxidation of phenolics by the action of polyphenol oxidase (PPO) enzyme (*Torres-Rodríguez et al., 2011*). As was previously reported, the higher phenolic and flavonoid contents were found in leaves, than peels, and the least was for the flesh, where values detected were at $14.15 \pm 0.48$, $1.23 \pm 0.06$, and $0.73 \pm 0.1$ µg GAE/100 g, respectively, for 70% ethanol extract of each organ (*Tamsir et al., 2020*).

### *In vitro* antioxidant assays

Assessment of the antioxidant activity of sapota fruit pulp extract was carried out using DPPH, ABTS scavenging assays, and FRAP assay to estimate its reducing property. Results revealed moderate effects at $1.62 \pm 0.2$, $1.49 \pm 0.11$, and $3.58 \pm 0.14$ mg TE/g extract as per DPPH, ABTS, and FRAP assays, respectively. According to previous reports, the highest antioxidant activity was exhibited by leaf ($92.96 \pm 0.06\%$), then peel ($91.98 \pm 0.71\%$), much higher than that of fruit pulp ($78.21 \pm 0.04\%$ of DPPH scavenging activity) as was reported in this study (*Tamsir et al., 2020*).

Metabolites profiling of fruit pulp showed that sphingoid bases and fatty acyl amides were the most abundant components, to likely contribute to the antioxidant activities. Prior studies proved that sphinganine inhibits the transport of cholesterol and low-density lipoprotein (*Tamsir et al., 2020*; *Roff et al., 1991*). Furthermore, previous findings confirm that monounsaturated fatty acids regulate several biochemical events (GABA, cannabinoid and anabolic pathways) within the cells (*Murphy, 2015*; *Divito & Cascio, 2013*). Additionally, other constituents may work together synergistically to boost antioxidant effectiveness (*Kumar et al., 2022*). Herein, several metabolites detected in the present study were reported for their antioxidant effect, including sugar alcohols, *viz*, mannitol (*Kang et al., 2007*), allyl isothiocyanates (*Caglayan et al., 2019*), palmitic acid, linoleic acid, (*Henry et al., 2002*), organic acids, *viz*, malic acid (*Gąsecka et al., 2018*), along with anethole (*Aprotosoaie et al., 2019*), curlone (*Jayaprakasha et al., 2002*), limonene (*El Omari et al., 2023*), cineole (*Hoch et al., 2023*).

### *In vitro* enzymes inhibition assays

Fruit pulp extract was assessed for its hypolipidemic and antidiabetic activities *via in vitro* assays targeting the inhibition of pancreatic lipase (PL) and $\alpha$-glucosidase enzymes, respectively. PL inhibitory assay tested the extract's influence on enzymatic activity and its potential for obesity management and lipid metabolic disorders. Results revealed that sapota fruit extract inhibited lipase enzyme by $IC_{50} = 4.42 \pm 0.5$ mg/mL and $2.21 \pm 0.25$ mg/mL at sample concentrations of 10 and 5 mg/mL, respectively, compared to the standard drug, Orlistat which showed $IC_{50}$ values of 0.16 and 0.08 mg/mL, Table 4.

Lipase enzyme plays a major role in fat metabolism. its downregulation leads to a decrease in low-density lipoprotein (LDL) and an increase in high-density lipoprotein

**Table 4 Enzymes inhibitory actions of sapota fruit extract, at 2 concentrations, compared to the positive controls.**

| IC$_{50}$ (mg/mL) | Pancreatic Lipase (PL) Inhibition Assay | | $\alpha$-Glucosidase inhibitory Assay | |
|---|---|---|---|---|
| Sample Conc. | SF ext. | Orlistat | SF ext. | Acarbose |
| 10 mg/mL | $4.42 \pm 0.5$ | 0.16 | NA | 0.5 |
| 5 mg/mL | $2.21 \pm 0.25$ | 0.08 | NA | 0.16 |

(HDL) (*Liu et al., 2020*), and provides health benefits for obesity prevention, management, and its related disorders (*Marzouk et al., 2024*). In the current study, the major metabolites detected in sapota fruit included sphingolipids, fatty acyl amides, and phospholipids, which could relate to its potential lipase inhibitory effect.

Previous studies reported the efficacy of dietary sphingolipids as a highly effective nutritional aid in improving metabolic syndromes and their outcoming disease, including, atherosclerosis, obesity, type 2 diabetes mellitus, and non-alcoholic fatty liver disease (*Wang et al., 2021*). Additionally, supplementation of phytosphingolipids was found to lower plasma triglycerides, low-density lipoprotein cholesterol levels, and improve clearance of glucose (*Snel et al., 2010*). *Liu et al. (2015)* reported that dietary sphingolipids effectively lowered epididymal adipose tissue weights, inhibited hepatic triglycerides levels, and serum glucose, and suppressed lipid uptake, moreover, increased the rate of triglycerides catabolism.

Fatty acyl amide is involved in the metabolic homeostasis of the human system (*Tanvir, Javeed & Rehman, 2018*). Likewise, phospholipids, amphiphilic lipids rich in sapota pulp, have been implicated in exhibiting a favored impact on blood lipids by reducing TG, total cholesterol, and LDL levels (*Küllenberg et al., 2012*). Moreover, terpenoids detected by GC/MS are reported as pancreatic lipase inhibitors (*Singh et al., 2015*). Compared to the potential lipase inhibition effect in fruit pulp. There is no effect observed regarding the $\alpha$-glucosidase inhibitory action. It was found that fruits were inactive against the enzyme compared to the positive drug control acarbose. This finding could be attributed to the decline in phenolic constituents and increased free sugars upon fruit maturation.

## CONCLUSION

This study reports on the metabolic profile of sapota fruit pulp *via* UPLC/MS and GC/MS techniques. SPME-GC/MS analysis revealed 17 aroma compounds to account for fruit aroma. Concerning nutrient composition, GC/MS analysis revealed 68 peaks belonging primarily to sugars accounting for fruit sweetness, and high-calorie content. Thirty-one metabolites were annotated using HR-UPLC/MS/MS which are reported for the first time in sapota fruit. For standardization of fruit pulp in terms of its total phenolics and flavonoids, a moderate level of phenolics was detected, while no flavonoids were noted. The antioxidant assays revealed a moderate free radical scavenging effect *via* DPPH and ABTS assays, *versus* moderate reducing capacity by FRAP assay. Fruit pulp extract exerted a considerable pancreatic lipase inhibitory (PL) action *versus* no $a$-glucosidase inhibition effect likely attributed to moderate levels of phenolics and absence of flavonoids. The

current study presents a comprehensive profiling of phytochemicals to provide better insight into sapota fruit's nutritive and health benefits. Future research is recommended to identify the best extraction solvents targeting the recovery and yield of its phytochemicals. Also, the promising lipase inhibitory action of fruit pulp motivates investigating various extracts to determine the most effective one for identifying natural antiobesity agents of natural origin.

## ACKNOWLEDGEMENTS

The authors are very grateful to Dr. Rupesh K. Deshmukh. Central University of Haryana, India for his kind help in the collection and authentication of sapota fruits.

### Funding
The authors received no funding for this work.

### Competing Interests
Mohamed A Farag is an Academic Editor for PeerJ.

### Author Contributions
- Mohamed A. Farag conceived and designed the experiments, performed the experiments, analyzed the data, prepared figures and/or tables, authored or reviewed drafts of the article, and approved the final draft.
- Nermin Ahmed Ragab performed the experiments, analyzed the data, prepared figures and/or tables, and approved the final draft.
- Maii Abdelnaby Ismail Maamoun performed the experiments, analyzed the data, prepared figures and/or tables, and approved the final draft.

### Data Availability
Raw data are available in the Supplemental Files.

### Supplemental Information
Supplemental information for this article can be found online at http://dx.doi.org/10.7717/peerj.17914#supplemental-information.

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
