# Peer review of "Metabolites profiling of Sapota fruit pulp via a multiplex approach of gas and ultra performance liquid chromatography/mass spectroscopy in relation to its lipase inhibition effect"

_PeerJ, doi:10.7717/peerj.17914_

## Round 0.1 · original submission · Major Revisions

Please address all the issues raised by the reviewers.

·

Basic reporting

The article comes in the scope of this reputed journal. However, some necessary corrections are required as highlighted through track change in MS- Word manuscript file. The abstract need revision to make it more attractive as it the most read part of manuscript. The introduction on plant is lengthy. Focus is required on the addition of alpha-glucosidase and pancreatic lipase inhibition to manage diabetes and obesity in introduction. The contents related to oxidative stress and role of natural antioxidants are also missing in introduction. Similarly, the toxicity of existing anti-obesity and antidiabetic agents and synthetic antioxidants must be discussed in introduction and discussion portion. Some sentences need rephrasing as already mentioned through track change. The article needs necessary revisions. After that the article may be published if adhere to the journal policies.
In the end of introduction, the need of current work must be established on sound grounds.

Experimental design

1. The work is original and comes under the scope of the journal.
2. All the required tests were performed to test the hypothesis. The analytical techniques used for metabolite identifications are latest and provide substantial information.
3. The extract yields were not calculated and only a single solvent system was used for extraction which may limit the actual potential of study.
4. The methods under section 2.8, 2.9 and 2.10 need revision in detail as they are not reproducible.
5. The data was not sufficient for the statistics application.
6. The DPPH assay IC50 value using trolox was unexplained. It needs some additional information which will be helping for the readers.

Validity of the findings

1. Information on identification of secondary metabolites using GC-MS and LC-MS/MS added novel information in existing pool of knowledge.
2. The libraries and sources used for metabolite identification including literature were upto the mark.
3. Discussion portion regarding lipase inhibition needs some revision. The molecular level changes related to sphingolipids modulations as obesity factor must be discussed in detail using suitable literature.
4. The reason having no alpha-glucosidase activity by fruit pulp must be clearly elaborated in discussion section.
5. Conclusion is too long, need to short it.
6. Overall, the results were informative and corresponded to the hypothesis or study question.

Additional comments

The manuscript need revision and changes mentioned above and in the manuscript file may be made to make the article more attractive and user friendly. Some grammatic and typo errors were also mentioned to be addressed.

Reviewer 2 ·

Basic reporting

The aim of this research is unclear. The introduction should include the benefit of metabolomic methods for revealing the nutritive property of agroproducts, and the novelty of this research should be highlighted.

Experimental design

no

Validity of the findings

The research seems to just identify some chemicals and evaluate the TPC, antioxidant capacity, and inhibitory effect of pancreatic lipase (PL) and alpha-glucosidase. But the significance and novelty was not clearly stated.

Additional comments

1. The relation between the results of metabolites profiling and inhibitory effect of pancreatic lipase and alpha-glucosidase enzymes were quite weak, and no inhibitory effect towards alpha-glucosidase was detected according to the results. This makes the title confusing.
2. Line 148: SPME analysis should be SPME-GS/MS analysis.
3. Line 227: Figure 2 should be Figure 1.
4. Figure 1, 2: What does the percent (10%-90%) in the y-axis mean?
5. Several references were not listed in a correct style.

·

Basic reporting

English language should be improved to make the article understandable internationally. The language may be improved including lines 51,55,79,108,211 and 230 as identified in review. Manuscript overall needs to be improved in well versed language format.

Experimental design

Introduction lacking background knowledge regarding medicinal benefits of plant extract hence needs to be well explained and specific. Extra details of plant may be reduced. In methodology section, invitro antioxidant assay, TPC and TFC needs to be well explained scientifically.

Validity of the findings

Findings are informative and addition to existing knowledge of metabolite profiling in medicinal research.

---

## Round 0.2 · Minor Revisions

Dear Authors

Thank you for addressing the issues raised by our reviewers. There are still some minor issues as highlighted by one of our reviewers in addition to some comments by myself as follows below:

The plant material was collected and lyophilized in India but no Indian co-author is there in the author list. How did you collect the sample?

The amount of Pancreatic lipase is mentioned as (5mg/ml) which is not a correct way of expressing the enzyme quantity. Please mention U/mL. Same is the case with alpha glucosidase. Name the substrates used for their activity. A brief elaboration of the enzyme activity may please be given. The readings in Table 4 are not self explanatory. No units are given.

English language still needs to be improved. There are still a number of grammatical errors in the text. Attention may also be paid towards paragraphing, there are some paragraphs comprising of single sentence.

Conclusion needs to be re-phrased and a very brief conclusion may please be given which actually reflects the outcome of the study.

You are requested to address all these queries and re-submit you MS for further evaluation.

·

Basic reporting

The required changes have been incorporated.

Experimental design

Necessary changes have been considered by the authors.

Validity of the findings

Seems fine.

Additional comments

Article may be published.

Reviewer 2 ·

Basic reporting

no comment

Experimental design

no comment

Validity of the findings

no comment

Additional comments

1. line 94: solid phase micro-extraction should be abbreviated as SPME, but not SPME-GC/MS. Similar error exsists in line 120.
2. According to the response, the y-axis in Figure 1 and Figure 2 represent the peak intensity. However, it is confusing to show the data as percent.

·

Basic reporting

No comments

Experimental design

No comments

Validity of the findings

No comments

Additional comments

Satisfactory

---

## Round 0.3 · accepted · Accept

All the queries have been resolved.